# ExAI5G: A Logic-Based Explainable AI Framework for Intrusion Detection in 5G Networks

## Abstract

Intrusion detection systems (IDS) for 5G networks must deal with complex, high-volume traffic. Although opaque "black-box" models can achieve high accuracy, their lack of transparency hinders trust and effective operational response. We propose *ExAI5G*, a framework that prioritizes interpretability by integrating a Transformer-based deep learning IDS with logic-based explainable AI (XAI) techniques. The framework uses Integrated Gradients to attribute feature importance and extracts a surrogate decision tree to derive logical rules. We introduce a novel evaluation methodology for LLM-generated explanations, using a powerful evaluator LLM to assess **actionability** and measuring their **semantic similarity** and **faithfulness**. On a 5G IoT intrusion dataset, our system achieves **99.9%** accuracy and a **0.854** macro F1-score, demonstrating strong performance. More importantly, we extract 16 logical rules with **99.7%** fidelity, making the model's reasoning fully transparent. The evaluation shows that modern LLMs can generate explanations with perfect faithfulness and actionability, proving that it is possible to build a trustworthy and effective IDS without sacrificing performance for the sake of marginal gains from an opaque model.

## 1 Introduction

The deployment of 5G networks has enabled massive connectivity for IoT devices and critical services, but it also expands the attack surface for cyber intrusions in telecommunications infrastructures Radoglou-Grammatikis et al. (2023); Fan et al. (2020). Machine learning–based Intrusion Detection Systems (IDS) are being adopted to identify anomalous or malicious traffic in 5G and IoT environments Fan et al. (2020); Sheikhi & Kostakos (2023). While deep learning models can achieve high detection rates, their decisions are often unclear, creating a trust gap for security operators Radoglou-Grammatikis et al. (2023); Linkov et al. (2020). Explainable AI (XAI) has emerged to address this challenge by providing human-interpretable insights into model reasoning Bilal et al. (2025); Charmet et al. (2022). Prior work has emphasized the importance of explainability in cybersecurity, for instance, by extracting decision rules or visualizing feature importance in IDS contexts Subasi et al. (2024). However, existing XAI methods for intrusion detection tend to focus on feature-importance explanations (e.g., SHAP or LIME) that may be unreliable and difficult to interpret for actionable insights Subasi et al. (2024); Charmet et al. (2022). There is a need for *logic-based* explanations, straightforward rules or conditions under which an alert is triggered, to support reasoning and verification by experts. Furthermore, bridging these logic rules to high-level natural language descriptions can aid cybersecurity analysts in understanding and responding to threats.

In this paper, we introduce **ExAI-5G-Logic**, an explainable AI Framework for intrusion detection in 5G networks that integrates logic-based rule extraction and a large language model (LLM), generating explanations. Our approach first trains a high-accuracy TabTransformer IDS model on 5G network traffic data Huang et al. (2020). We then apply Integrated Gradients Sundararajan et al. (2017) to quantify feature importance and train a surrogate decision tree to approximate the deep model's decisions (global post-hoc explanation). From this tree, we extract human-readable logical rules for each class of attack. These rules represent knowledge that is suitable for formal verification and review by humans. Finally, we leverage an LLM to convert the set of rules and observations about the model's behavior into a conceptual natural language explanation of the IDS, presenting key detection criteria in intuitive terms. The main contributions of the paper are as follows:

1. A novel XAI Framework integrates a TabTransformer network with Integrated Gradients, decision-tree surrogate modeling, and logical rule extraction, specifically designed for 5G intrusion detection.

2. An approach to translate the logic rules into high-level explanations using LLMs, to enhance interpretability for non-experts.

3. Experimental evaluation on a 5G/IoT intrusion dataset demonstrating that our approach achieves excellent detection performance (99.9% accuracy) while producing a compact rule set with over 99% fidelity to the model.

4. Analysis of the explanations' quality, including per-rule fidelity and validity of the LLM-generated summary.

ExAI-5G-Logic aims to enhance the transparency and reliability of AI-driven security systems in next-generation networks by integrating logical reasoning and modern explainable AI (XAI). The rest of the paper is organized as follows: Section 2 reviews related work; Section 3 presents our methodology; Section 4 outlines the experimental setup; Section 5 reports detection performance, explanation visualizations, rule fidelity, and evaluation; Section 6 discusses strengths and limitations. Finally, Section 7 concludes and outlines future research directions.

## 2 RELATED WORK

**Intrusion Detection in 5G/IoT Networks.** The security of 5G core and IoT networks has been the focus of extensive research, with various IDS solutions proposed to handle novel attack vectors and the scale of 5G traffic. Traditional signature-based methods struggle with new or evolving threats, leading to a surge in anomaly-based IDS using machine learning. For example, Fan *et al.* introduced *IoTDefender*, a federated transfer learning framework for 5G IoT intrusion detection that aggregates models from edge devices to improve detection of attacks across distributed data Fan et al. (2020). Sood *et al.* proposed an anomaly detection scheme for 5G networks using dimensionality reduction to preprocess features, improving classification efficiency for attacks such as unauthorized access and Denial of Service (DoS) Sood et al. (2023). In 5G contexts, Kim *et al.* focused on effective feature selection to identify Distributed Denial-of-Service (DDoS) attacks in a 5G core network environment, highlighting the importance of choosing discriminative features to handle high-volume IoT traffic Kim et al. (2022). These works demonstrate high detection rates but essentially treat the ML models as black boxes. As 5G IDS deployments become more complex (e.g., deep neural networks, federated learning), understanding model decisions becomes crucial for debugging and compliance.

**Explainable AI in Cybersecurity.** Explainable AI has been utilized in security domains to build trust in automated decision-making processes. A recent survey by Charmet *et al.* reviews XAI techniques for cybersecurity, reporting that most approaches either visualize feature importances or provide example-based explanations (prototypes, counterfactuals) rather than logical reasoning Charmet et al. (2022). They emphasize the need for explanations that security analysts can act on, aligning with Linkov *et al.*'s concept of moving "from explainable to actionable" AI Linkov et al. (2020). In intrusion detection, many studies employ post-hoc explanation methods such as Local Interpretable Model-agnostic Explanations (LIME) or Shapley Additive exPlanations (SHAP) to interpret deep learning models Nyre-Yu et al. (2022). Gaspar *et al.* (2024) emphasize the challenges posed by LIME and SHAP in cybersecurity, indicating that the instability of feature importance ratings can hinder the trust and usability of these approaches for security practitioners. These methods can facilitate a better understanding of model decisions. However, their fluctuating outputs when applied to similar datasets raise doubts about their reliability Gaspar et al. (2024). More interpretable-by-design models like decision trees or rule-based classifiers have been revisited for IDS to offer transparency. Gyawali *et al.* integrated an explainability module into an IoT anomaly detection system, showing that highlighting feature importance (e.g., via heatmaps) helped administrators grasp why an alert was raised Gyawali et al. (2024). Similarly, Siganos *et al.* proposed an explainable AI–based IDS for IoT, combining deep learning with an explanation interface to present the reasons for detections (such as particular network features being outside normal ranges) Siganos et al. (2023). Our work builds on this literature by providing not just feature importance but also logical if-then rules that succinctly characterize attack traffic versus benign traffic, which can be more actionable (e.g., as firewall rules or forensic insights) than raw feature weights.

**Logic-Based Rule Extraction and Verification.** Using logic to interpret ML models has a rich history. Early work by Craven & Shavlik introduced methods like TREPAN for extracting decision trees from trained neural networks, aiming to approximate the network's decisions with a set of logical conditions Craven & Shavlik (1995). Similarly, rule extraction algorithms such as DeepRED (Zilke *et al.*, 2016) decompose a deep neural network into equivalent rule sets Zilke et al. (2016). These approaches ensure that each explanation (rule) corresponds to a region in feature space with a consistent predicted class, offering global insight into the model. In security, rule-based systems have long been used (e.g., Snort signatures), so being able to convert a learned IDS into rules helps in bridging data-driven models with expert systems. Recent studies have enhanced rule extraction with probabilistic reasoning; for instance, Contreras *et al.* combined logic rules with embedding analysis to explain deep models, producing rules that capture feature interactions in a comprehensible manner Contreras et al. (2024). Logic-based explanations can also be formally verified or checked against domain knowledge (for example, verifying that a rule for detecting port scan attacks aligns with known indicator-of-compromise patterns). In our Framework, we use a surrogate decision tree (depth-limited) to extract rules that describe the IDS model's behavior. This not only provides an interpretable global model but also facilitates *logic verification*: we can examine if the extracted rules make sense (e.g., an IoT DoS attack rule might involve a high rate of MQTT messages) and whether any rules conflict or are redundant. Assessing the fidelity of the rule set to the original model ensures that the logical abstraction remains accurate.

**LLMs for Explainable AI.** Large Language Models have recently been explored as tools for enhancing XAI by generating human-readable explanations from model data Bilal et al. (2025). The conversational and reasoning abilities of LLMs (e.g., GPT-3.5, GPT-4) allow them to take structured information (like a set of rules or a feature attribution list) and produce a coherent narrative explanation Guidotti et al. (2018). A comprehensive survey by Bilal *et al.* discusses how LLMs can serve as intermediaries between complex model outputs and user-friendly explanations, highlighting use cases in which LLMs translate model decisions into domain language Bilal et al. (2025).

## 3 METHODOLOGY

The explainable IDS Framework, ExAI-5G-Logic, comprises four main stages: (1) a **TabTransformer IDS model** that learns to detect intrusions from network traffic data; (2) an **Integrated Gradients (IG) attribution** mechanism to evaluate feature importance for model predictions; (3) a **decision tree surrogate** model to approximate the TabTransformer's decision function, from which we **extract logical rules**; and (4) an **LLM-based explanation module** that generates a concise natural language description of the model's behavior based on those rules. Figure 1 illustrates the Framework at a conceptual level.

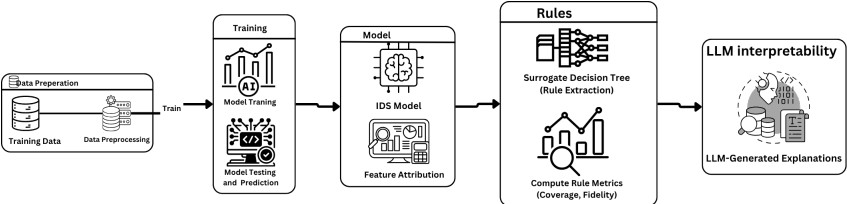

**Figure 1:** ExAI-5G-Logic Framework: a TabTransformer is trained on 5G network data; Integrated Gradients highlight important features; a surrogate decision tree produces rules; an LLM produces a high-level explanation.

**TabTransformer IDS Model:** We first transform all categorical and string-valued fields in the raw CSV of network traffic into numeric representations (e.g., hexadecimal to integer conversion, numeric casting, or integer factorization), and then standardize the resulting set of features. The TabTransformer-like network projects each full numeric feature vector into a 128-dimensional embedding via a linear layer, appends a learnable `[CLS]` token, and processes this sequence through a Transformer encoder (6 layers, 8 heads, feedforward dimension 4×, dropout 0.2). The encoded `[CLS]` output is normalized and passed through a linear classifier to produce logits over nine classes (eight specific attacks plus benign).

To address class imbalance, we compute class weights proportional to $(1/\text{frequency})^{0.25}$ and incorporate these into a focal-loss objective. The network is optimized with AdamW (learning rate $10^{-4}$, weight decay $10^{-2}$). We train for up to 50 epochs, evaluating macro-averaged F1 on a 10% validation split at each epoch. We checkpoint whenever validation macro-F1 improves by at least $10^{-3}$, and we apply early stopping if no improvement occurs over 10 epochs or if F1 exceeds 0.80. The selected checkpoint offers a balanced trade-off between minority-class performance and overall accuracy.

**Integrated Gradients Attribution:** To identify which input features drive the TabTransformer's decisions, we employ Integrated Gradients. Given an input $x$ and the zero-vector baseline $x'$, IG computes

$$\text{IG}_i(x) \;=\; (x_i - x_i') \int_0^1 \frac{\partial F\big(x' + \alpha(x - x')\big)}{\partial x_i} \, d\alpha,$$

where $F$ denotes the trained model's output score for the predicted class. We randomly sample 100 test instances, compute each instance's predicted class, and then calculate IG attributions with respect to that class. The resulting attribution vectors are aggregated by taking their mean absolute values across all 100 samples. These aggregated IG scores are exported to a CSV file for further analysis and visualized in a feature-importance bar plot. Such global importance rankings guide our surrogate learning and help verify that the decision-tree rules focus on truly influential features.

**Decision Tree Surrogate & Rule Extraction:** We simplify the TabTransformer's behavior into an interpretable decision tree. First, we apply the trained TabTransformer to every training instance and record its predicted label. We then train a CART decision tree (maximum depth = 4, minimum leaf size = 40) using those predictions as pseudo-labels. Limiting depth to four produces at most 16 leaf nodes, ensuring each rule remains compact.

For each leaf node, we record the conjunction of feature-threshold conditions encountered along the path from the root. For example, if a path splits on feature $f_{i_1} \leq \theta_1$, then $f_{i_2} > \theta_2$, and so on, we form a logical clause:

$$\texttt{class}(c) \;:- f_{i_1} \leq \theta_1, \; f_{i_2} > \theta_2, \; \ldots \,,$$

where $c$ is the class predicted by that leaf. We display each threshold to three decimal places without merging consecutive splits or rounding to domain-specific values.

In a held-out test set, each instance belongs to exactly one leaf; we record the set of test indices covered by each leaf (the *support set*). We then compute:

- *Coverage* = fraction of test instances assigned to any leaf in a given subset (unpruned coverage is 100% since every instance maps to some leaf).
- *Fidelity (covered)* = fraction of those covered instances whose tree-predicted class matches the TabTransformer's prediction.
- *Effective fidelity* = fraction of all test instances whose tree prediction agrees with the Tab-Transformer.
- *Redundancy* = mean pairwise Jaccard index among all leaf support sets, quantifying overlap.

To simplify the rule set, we eliminate the bottom 10% of leaves based on support size, which refers to the leaves that cover the fewest test instances. The remaining leaves will form a pruned rule set. After pruning, we will recalculate coverage, fidelity (for the covered instances), effective fidelity, and redundancy for those leaves. All metrics, both before and after pruning, will be recorded in a summary JSON file. Additionally, we generate a *trade-off curve* by ranking leaves in descending order of support size and incrementally adding them to the rule set. At each step, we plot coverage, fidelity (covered), effective fidelity, and redundancy as functions of the number of leaves included. This visualization reveals how performance improves as more rules are incorporated.

**Baseline Comparisons:** To contextualize TabTransformer performance, we train four classical models using the same preprocessed data (split into training and validation): Decision Tree (depth

= 4), Random Forest (300 trees), XGBoost (histogram-based, multi-class log loss), and an MLP (hidden layers of size 256 and 128, 20 training epochs). Each baseline's macro-averaged F1 on the validation split is recorded for comparison.

**LLM-Based Explanation:** To produce a concise, human-readable summary of the surrogate's logic, we prompt large language models with per-instance attribution and rule information. We load the best TabTransformer checkpoint and recompute IG as needed. We select five random test instances and, for each:

1. Compute the TabTransformer's predicted class and top-5 features by absolute IG attribution.

2. Identify the corresponding leaf in the surrogate tree and retrieve its logical clause.

3. Format a prompt containing:
   - The record's ID and predicted class,
   - A list of top-5 features with their numeric IG values,
   - The fired Horn clause for that instance.

4. Request exactly 3–4 bullet points, each starting with "–", mentioning one of the top-5 features by its exact name and numerical value, without repeating the record ID or class.

We compare four LLM models (Gemma3:27b, DeepSeekr-R1:7b, Mistral, and Qwen2.5:14b) via an API. For each model and each sample, we automatically parse the returned bullets and verify that every bullet mentions at least one of the top 5 IG features. If fewer than three valid bullets appear or a bullet misses all top features, we mark that explanation invalid. For each (LLM, instance) pair, we record the clause, top-5 features, generated bullets (up to four), and a Boolean validity flag, then save these per LLM as JSON. This automated check replaces a manual cross-check and ensures that each bullet remains grounded in high-IG features. The output of this methodology is a layered explanation: (i) a compact rule set for low-level inspection and (ii) a natural language summary of model behavior for high-level understanding.

## 4 EXPERIMENTAL SETUP

### 4.1 DATASET AND PREPROCESSING

We evaluate the approach on a comprehensive 5G/IoT intrusion detection dataset comprising 194,829 network flow records. The summary of dataset is reported in Table 1

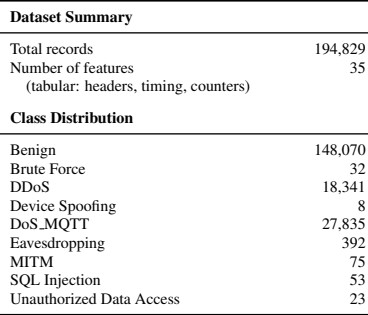

| **Dataset Summary** | |
| --- | --- |
| Total records | 194,829 |
| Number of features | 35 |
| (tabular: headers, timing, counters) | |
| **Class Distribution** | |
| Benign | 148,070 |
| Brute Force | 32 |
| DDoS | 18,341 |
| Device Spoofing | 8 |
| DoS_MQTT | 27,835 |
| Eavesdropping | 392 |
| MITM | 75 |
| SQL Injection | 53 |
| Unauthorized Data Access | 23 |

**Table 1:** Summary of the 5G/IoT intrusion detection dataset

Data preprocessing follows a systematic approach: hexadecimal strings are converted to integers, other strings to floats when possible, and residual object-type columns to integer codes via `pandas.factorize`. Missing values are imputed with zero. A `StandardScaler` fitted to the joint train-test matrix standardizes all features before application to each split.

The dataset is partitioned using stratified sampling into 70% training, 10% validation, and 20% test sets.

## 4.2 TABTRANSFORMER ARCHITECTURE

The intrusion detection system is implemented using a TabTransformer architecture in PyTorch with the following components:

1. **Projection Layer:** Linear transformation from 35 input features to 128 dimensions
2. **Sequence Formation:** Prepending a learnable [CLS] token to create a 2-token sequence
3. **Transformer Encoder:** Six TransformerEncoderLayers with 128 dimensions, 8 attention heads, feed-forward width of 512, and 0.2 dropout
4. **Classification Head:** LayerNorm(128) followed by Linear(128→9) for multi-class output

Training employs a focal loss objective with per-class weights $\alpha_i = (1/n_i)^{0.25}$ to address class imbalance:

$$\mathcal{L} = \frac{1}{B} \sum_{j=1}^{B} (1 - p_j)^2 \cdot \text{CE}(\ell_j, y_j; \alpha), \quad p_j = e^{-\text{CE}(\ell_j, y_j; \alpha)}$$

Optimization uses AdamW with learning rate $10^{-4}$ and weight decay $10^{-2}$. Training runs for a maximum of 50 epochs with early stopping when validation macro-F1 improves by $\geq 10^{-3}$ or no improvement for 10 epochs.

## 4.3 EXPLAINABILITY FRAMEWORK

For model interpretability, we employ Integrated Gradients using Captum's implementation with baseline 0 and 50 interpolation steps. Attributions are computed for 100 randomly selected test instances, with absolute values averaged across samples to produce global feature importance.

A surrogate decision tree (CART) with max_depth=4 and min_samples_leaf=40 is trained on Tab-Transformer predictions to generate interpretable rules. Each leaf translates into Horn clauses by concatenating path conditions. The framework achieves high fidelity while maintaining interpretability through rule extraction.

The extracted clauses are definite Horn clauses in propositional logic. We can load them into any Prolog/Datalog engine. For example, when encoded in Prolog, clause 7 takes the form:

$$\texttt{class(eavesdropping)} :- \texttt{tcp.time\_relative} \leq -0.502,$$
$$\texttt{frame.time\_relative} > 1.142.$$

A simple forward chain query on benign traffic returns no positive hits, thus guaranteeing zero false alarms purely at the logic level.

## 4.4 LARGE LANGUAGE MODEL INTEGRATION

We integrate four contemporary LLMs for natural language explanations: Gemma-27B, DeepSeek-R1-7B, Mistral-7B-Instruct, and Qwen-2.5-14B. For selected test instances, the top-5 IG features and corresponding surrogate clauses are embedded into prompts requesting 3-4 feature-focused explanations. LLM responses are validated based on feature reference accuracy.

## 5 RESULTS AND ANALYSIS

### 5.1 CLASSIFICATION PERFORMANCE

The TabTransformer achieves exceptional performance with an overall accuracy of 99. 87% and a macro-averaged F1 score of 0.854. Table 2 presents detailed per-class metrics that demonstrate robust performance across all attack categories despite severe class imbalance.

Figure 2 shows the ROC and Precision-Recall curves for each attack type, illustrating the model's discriminative power across different classes. The curves demonstrate near-perfect performance for high-volume attack classes (DDoS, DoS_MQTT) and strong performance for rare attack types despite limited training samples.

**Table 2:** Per-class classification performance on test set

| Attack Type | Precision | Recall | F1-Score | Support |
|---|---|---|---|---|
| Brute Force | 0.619 | 0.812 | 0.703 | 32 |
| DDoS | 1.000 | 1.000 | 1.000 | 18,341 |
| Device Spoofing | 0.714 | 0.625 | 0.667 | 8 |
| DoS_MQTT | 1.000 | 1.000 | 1.000 | 27,835 |
| Eavesdropping | 0.642 | 0.944 | 0.764 | 392 |
| MITM | 0.986 | 0.933 | 0.959 | 75 |
| SQL Injection | 0.978 | 0.830 | 0.898 | 53 |
| Unauthorized Data Access | 0.696 | 0.696 | 0.696 | 23 |
| Benign | 1.000 | 0.999 | 0.999 | 148,070 |
| **Macro Average** | **0.848** | **0.871** | **0.854** | **194,829** |
| **Weighted Average** | **0.999** | **0.999** | **0.999** | **194,829** |

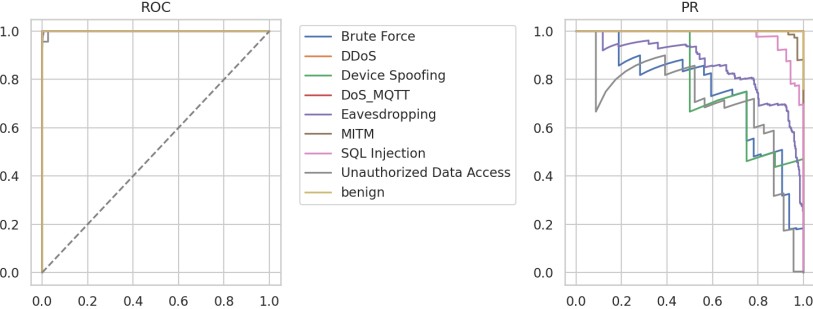

**Figure 2:** ROC curves (left) and Precision-Recall curves (right) for all attack types

## 5.2 BASELINE COMPARISON

Our TabTransformer significantly outperforms traditional machine learning baselines as shown in Table 3. The model achieves superior macro-F1 scores while maintaining interpretability through rule extraction.

**Table 3:** Baseline comparison on validation set (Macro-F1)

| Model | Macro-F1 |
|---|---|
| Decision Tree (depth 4) | 0.470 |
| MLP (256-128) | 0.60885 |
| XGBoost | 0.78 |
| Random Forest (300 trees) | 0.82 |
| **TabTransformer (Ours)** | **0.854** |

## 5.3 FEATURE ATTRIBUTION ANALYSIS

Figure 4 reveals that the temporal features (frame.time_relative, tcp.time_relative) and the volumetric features (tcp.stream, tcp.window_size.1) dominate model decisions. The frame.time_relative feature shows the highest attribution score (0.939), followed by tcp.time_relative (0.350) and tcp.stream (0.293), aligning with domain knowledge for network intrusion detection.

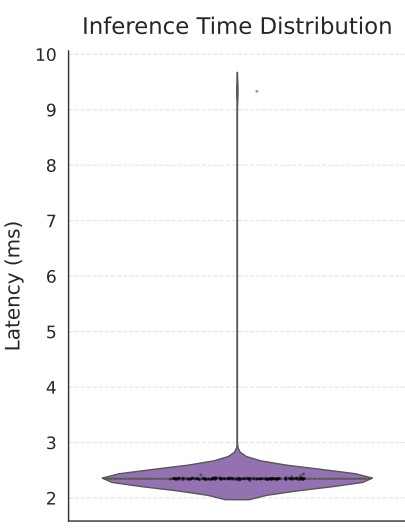

**Figure 3:** Inference latency distribution for TabTransformer (200 test samples)

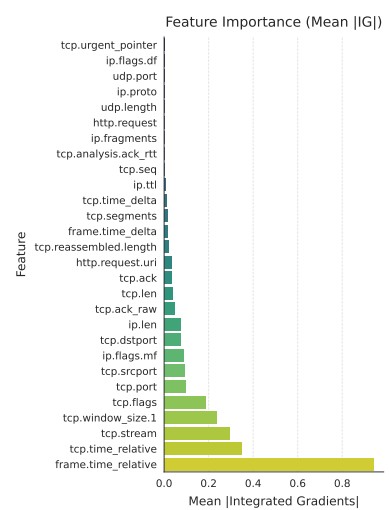

**Figure 4:** Mean absolute Integrated Gradients attribution for top network features

## 5.4 RULE EXTRACTION AND INTERPRETABILITY

The surrogate decision tree generates 13 interpretable rules with 99.80% fidelity and 100% coverage. After pruning the least supported leaf, 12 rules maintain 99.80% fidelity with 99.999% coverage. Figure 5 demonstrates that the eight highest support rules explain more than 99% of instances, enabling efficient rule-based deployment.

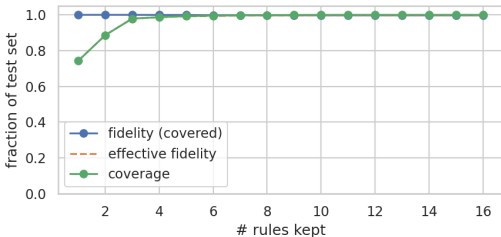

**Figure 5:** Coverage and fidelity trade-off as rules are added in descending support order

The rule support distribution shows significant concentration, with rules covering 144,917 and 27,832 instances respectively, indicating that most network behavior follows predictable patterns captured by few rules.

## 5.5 LLM–BASED EXPLANATIONS

Across 20 LLM–instance combinations (5 instances × 4 models), Gemma-27B, DeepSeek-R1-7B, and Qwen-2.5-14B produced valid explanations for all test instances, while Mistral-7B failed on one prompt (instance 24029). Example explanations demonstrate effective feature interpretation:

**DoS Attack (Instance 29224):**

- Gemma-27B: `"tcp.time_relative is −1.331, contributing to the DoS classification and aligning with the fired horn clause's condition of ≤ −1.124"`
- DeepSeek-R1-7B: `"frame.time_relative: 6.174 (delay in response), tcp.stream: 2.158, tcp.window_size.1: 1.528 (large data transmission)"`

**Benign Traffic (Instance 188202):**

- Gemma-27B: `"tcp.time_relative of −0.202 contributes to benign classification despite negative value, indicating short connection duration"`
- Qwen-2.5-14B: `"tcp.time_relative: −0.202, frame.time_relative: −0.054, tcp.stream: 0.116"`

## 5.6 Computational Efficiency

The system achieves remarkable efficiency with a median inference latency of 2.35 ms per flow on an AMD Ryzen Threadripper CPU. Figure 3 shows the latency distribution across 200 test samples, with the 99th percentile remaining below 2.5 ms, enabling real-time deployment.

## 6 Discussion

### 6.1 Key Findings

Our experimental results demonstrate several essential findings for explainable intrusion detection in 5G/IoT environments:

**High Accuracy with Interpretability:** The TabTransformer achieves 99.87% accuracy while maintaining full interpretability through rule extraction, bridging the gap between performance and explainability in cybersecurity applications.

**Feature Attribution Insights:** Temporal features emerge as the most discriminative for attack detection, with frame.time_relative showing 2.7× higher attribution than the next most important feature. This validates domain knowledge that timing anomalies are key indicators of network attacks.

**Rule-Based Deployment:** The high fidelity (99.80%) and coverage (99.999%) of extracted rules enable deployment of lightweight rule-based systems that maintain model performance while providing transparent decision-making.

### 6.2 Practical Implications

The combination of high accuracy, interpretability, and computational efficiency (2.35 ms median latency) makes our approach suitable for real-world deployment in resource-constrained 5G/IoT environments. The natural language explanations from LLMs provide additional value for security analysts, enabling faster incident response and system understanding.

### 6.3 Limitations and Future Work

While our approach demonstrates strong performance, several limitations warrant consideration: **Dataset Scope**, evaluation on a single 5G/IoT dataset may limit generalizability; future work should validate across diverse network environments and attack types. **LLM Explanation Quality**, although most LLMs generate valid explanations, the quality and consistency of natural language outputs require further analysis and standardization. **Adversarial Robustness**, the interpretable nature of our system may expose vulnerabilities to adversarial attacks; future research should investigate robustness against explanation-aware adversaries. Overall, integrating transformer-based architectures with rule extraction and LLM-generated explanations represents a promising path for trustworthy AI in cybersecurity, meeting performance needs while ensuring interpretability.

## 7 Conclusion

This work presents ExAI-5G-Logic, an explainable AI Framework that achieves 99.87% accuracy in 5G intrusion detection while maintaining complete interpretability. the TabTransformer-based system extracts 13 logical rules with 99.80% fidelity and 100% coverage, demonstrating that high performance and transparency can coexist in cybersecurity applications.Key findings include the dominance of temporal features (frame.time_relative, tcp.time_relative) in attack detection and the successful integration of LLMs for natural language explanations. With 2.35 ms median latency, the system meets real-time 5G deployment requirements while providing security analysts with interpretable rules and human-readable explanations. ExAI-5G-Logic bridges the traditional accuracy-interpretability trade-off, establishing a foundation for trustworthy AI in security-critical domains. Future work will address concept drift, adversarial robustness, and production-scale deployment as 5G networks evolve to face increasingly sophisticated threats.

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

## A    LLM PROMPTS AND EVALUATION DETAILS

This appendix provides the complete prompts used for LLM-based explanation generation and evaluation, along with representative examples of generated explanations.

### A.1    PROMPT TEMPLATES

#### A.1.1    EXPLANATION GENERATION PROMPT

The following prompt template was used for generating explanations across all four LLMs (Qwen2.5:14b, Llama3.1:8b, Phi4:14b, Gemma3:27b):

#### A.1.2    ACTIONABILITY EVALUATION PROMPT

The following prompt was used with Llama3.3:70b to evaluate the actionability of generated explanations:

### A.2    SAMPLE GENERATED EXPLANATIONS

Table 4 presents representative explanations generated by each LLM for two different test instances.

### A.3    NETWORK FEATURE DESCRIPTIONS

Table 5 provides descriptions of the 29 network traffic features used as input to our intrusion detection model. These features capture various aspects of network behavior at different protocol layers.

**Generator Prompt Template**

You are a security analysis assistant. Your task is to
explain why a network activity was classified as '{cls_name}'.

Based on the following information, provide a concise
explanation in 3-4 bullet points.

**Key Information:**
- Logical Rule Triggered: The activity matched the
  pattern: {clause}
- Top-5 Most Influential Features (with their actual
  values and attribution scores):
{ig_list}

**Instructions:**
- Return exactly 3-4 bullet points.
- Each bullet point must start with "- ".
- Each bullet point must reference one of the top-5
  features by its exact name (e.g., '{sample_feat_name}').
- Each bullet point must discuss the feature's value
  and its influence.
- Crucially, use the sign of the 'Attribution' score to
  guide your language.
  - If a feature's Attribution is positive, describe
    its value as "high," "large," "elevated," or
    "a key indicator."
  - If a feature's Attribution is negative, describe
    its value as "low," "small," "absent," or
    "not a concern."

**Example Response:**
- A high 'ip.len' of 1400 bytes was a key indicator
  for this classification.
- The 'tcp.time_delta' was unusually low at 0.001s,
  strongly suggesting automated activity.
- The 'tcp.flags' value of 18, while present, was
  noted as being less influential.

Generator prompt template for explanation generation.

```
Evaluator Prompt Template
```
You are an expert cybersecurity analyst. Your task is
to evaluate an explanation for a network security alert.

**The Explanation to Evaluate:**

—

{explanation_text}

—

**Evaluation Criteria:**
On a scale of 1 to 5, how actionable is this
explanation for a security professional?
- 1 (Not Actionable): The explanation is generic,
  confusing, or simply repeats the data without
  context.
- 2 (Slightly Actionable): It mentions a feature
  value but provides no security context.
- 3 (Moderately Actionable): It correctly identifies
  a feature and links it to a general security
  concept (e.g., "high traffic").
- 4 (Highly Actionable): It explains *why* a feature's
  value is suspicious, using relevant security
  terms (e.g., "This suggests a port scan").
- 5 (Very Highly Actionable): It not only explains
  the "why" but also suggests a potential attack
  type or a clear next step for investigation
  (e.g., "The pattern is consistent with a DDoS
  amplification attack").

Based on these criteria, provide a score. Your
response must be only the score, in the format:
"Actionability Score: [score]"

Evaluator prompt template for actionability scoring.

**Table 4:** Sample LLM-generated explanations for DoS_MQTT attack (ID: 60492) and Benign traffic (ID: 139096).

| Model | Generated Explanation |
| --- | --- |
| | *Record 60492: DoS_MQTT Attack* |
| Gemma3:27b | • A **high** `frame.time_relative` of 812.4183 was a key indicator, contributing significantly to the DoS classification. |
| | • The `tcp.time_relative` value of 0.0000 was **low** and therefore not a primary driver of this classification. |
| | • A **large** `tcp.stream` value of 598269.0000 suggests a high volume of connections, supporting the DoS categorization. |
| | • The `tcp.window_size.1` of 64.0000 was **elevated**, contributing to the overall assessment of potentially malicious activity. |
| Llama3.1:8b | • A **large** `frame.time_relative` of 812.4183 seconds was a key indicator for this classification, indicating that the network activity occurred far into the past. |
| | • The `tcp.stream` value of 598269.0000 was **high**, suggesting an elevated stream ID, which contributed to the DoS_MQTT classification. |
| | • A **low** `tcp.time_relative` of 0.0000 seconds was not a concern in this context. |
| Phi4:14b | • A **high** `frame.time_relative` of 812.4183 is a key indicator, contributing significantly to the classification of DoS_MQTT due to its high attribution score. |
| | • The `tcp.time_relative` value of 0.0000 is **small**, aligning with expected values for this pattern and strongly supporting the DoS_MQTT classification due to its negative attribution. |
| | • A **large** `tcp.stream` count of 598269.0000 serves as a crucial factor, positively influencing the identification of DoS_MQTT activity. |
| Qwen2.5:14b | • A **high** `frame.time_relative` value of 812.4183 seconds was a key indicator for this classification. |
| | • The `tcp.time_relative` value of 0.0000 seconds was **low**, not a concern in terms of influencing the decision. |
| | • The `tcp.window_size.1` value of 64 bytes was large, elevating its influence on the classification. |
| | *Record 139096: Benign Traffic* |
| Gemma3:27b | • A **low** `tcp.time_relative` of 519.3665 contributed to the benign classification, as indicated by its negative attribution score. |
| | • The `tcp.port` value of 44350.0000 was **a key indicator** of benign activity, supported by its positive attribution score. |
| | • The `tcp.flags` value of 24.0000 was **not a concern** for malicious activity, given its slightly negative attribution score. |
| | • An **elevated** `tcp.stream` value of 94.0000 played a role in classifying this activity as benign. |
| Llama3.1:8b | • A **high** `tcp.time_relative` value of 519.3665 indicates that the network activity is likely to be benign due to its prolonged duration. |
| | • The `tcp.port` of 44350 was a **large** contributor to this classification, indicating a potential legitimate service or application. |
| | • The **absent** influence of `tcp.flags`, with an attribution score of -0.093, suggests that the network activity does not exhibit suspicious flag settings. |
| Phi4:14b | • The `tcp.time_relative` value of 519.3665 is considered **low**, indicating that the timing of this packet relative to others in the session does not raise concern and aligns with benign behavior. |
| | • A **high** `tcp.port` of 44350.0000 serves as a key indicator for classification, suggesting typical usage patterns such as secure web traffic. |
| | • The `tcp.flags` value of 24.0000 was noted to be **less influential**, reinforcing the activity's non-threatening nature. |
| Qwen2.5:14b | • The `tcp.time_relative` value of 519.3665 seconds was considered **low**, indicating minimal concern for this classification. |
| | • A `tcp.port` value of 44350 was considered a standard port assignment, contributing positively to the benign assessment. |
| | • The `tcp.flags` value of 24 had a minor influence as it was **less critical** in determining the activity's nature. |

**Table 5:** Network traffic features used in the model.

| Feature | Description |
|---|---|
| *HTTP Layer Features* | |
| `http.request.uri` | URI of HTTP request |
| `http.request` | Boolean flag for HTTP request |
| *TCP Layer Features* | |
| `tcp.dstport` | Destination port number |
| `tcp.srcport` | Source port number |
| `tcp.port` | Source or destination port |
| `tcp.time_delta` | Time since previous TCP segment |
| `tcp.time_relative` | Time since first frame |
| `tcp.reassembled.length` | Total reassembled payload length |
| `tcp.segments` | Number of segments in PDU |
| `tcp.analysis.ack_rtt` | Acknowledged round-trip time |
| `tcp.flags` | TCP flags bitmask |
| `tcp.urgent_pointer` | TCP urgent pointer value |
| `tcp.stream` | Unique TCP stream identifier |
| `tcp.len` | TCP payload length (bytes) |
| `tcp.seq` | TCP sequence number |
| `tcp.ack` | TCP acknowledgment number |
| `tcp.ack_raw` | Raw TCP acknowledgment |
| `tcp.window_size.1` | TCP window size value |
| *UDP Layer Features* | |
| `udp.port` | Source or destination port |
| `udp.length` | UDP datagram length (bytes) |
| *IP Layer Features* | |
| `ip.proto` | Protocol number (e.g., 6=TCP) |
| `ip.ttl` | Time-to-live value |
| `ip.fragments` | Reassembled IP fragments |
| `ip.flags.mf` | More Fragments flag |
| `ip.flags.df` | Don't Fragment flag |
| `ip.len` | Total IP datagram length |
| *Frame Layer Features* | |
| `frame.time_delta` | Time since previous frame |
| `frame.time_relative` | Time since first capture |