# OpenReview forum: "ExAI5G: A Logic-Based Explainable AI Framework for Intrusion Detection in 5G Networks"
_ICLR.cc/2026/Conference — ICLR 2026 Conference Withdrawn Submission_

### Official Review · Reviewer_aokX · 2025-10-26

**Soundness:** 2
**Presentation:** 3
**Contribution:** 2
**Rating:** 2
**Confidence:** 4

**Summary:**

- The paper proposes ExAI-5G-Logic, a 5G IDS that trains a TabTransformer, ranks features via Integrated Gradients, extracts compact logic rules from a surrogate CART tree, and uses an LLM to generate natural-language explanations.
- It reports 99.87% accuracy (macro-F1 0.854), 12–13 rules with ~99.8% fidelity and ~100% coverage, and millisecond-level inference latency.

**Strengths:**

- Clear end-to-end pipeline combining IG attributions, rule extraction, and LLM summarization suited to analyst workflows.
- Rule set is compact with high fidelity/coverage, aiding deployability and audit by an analyst that will increase the confidence and interpretability of the underlying IDS.

**Weaknesses:**

- The work frames supervised multiclass classification as “intrusion detection,” overlooking that practical IDS often rely on anomaly/unsupervised paradigms.
- The paper does not define a threat model (adversary goals, knowledge, and capabilities), limiting security relevance.
- Since they claim that they know all the features used by the defender's model, I am assuming that they have a white box threat model, which, in case, makes their threat model extremely weak and unrealistic.
-“Attribution” is used informally; it appears to mean mean |IG|, but the 2.7× claim needs a precise definition and denominator.
- They also need to show that their explanation method and underlying IDS have some relationship. I think they need to use Fidelity+/- methods, where fidelity+ measures that by removing or perturbing the important features, the classification changes to what degree, and fidelity- measures that by removing or perturbing the unimportant features, the classification to what degree.
- No qualitative, end-to-end analyst case study showing how rules + LLM bullets drive concrete response actions.
- LLM “actionability” scoring relies on an automated LLM evaluator, which risks circularity without human or blinded baselines.
- CART hyperparameters (max depth=4, min leaf=40) feel arbitrary.

**Questions:**

- What is the precise threat model (white/gray/black-box; targeted vs untargeted; allowed perturbations; control over features)?
- Can you add a short appendix case study walking through one alert: packet fields → rule fired → LLM bullets → concrete analyst action?
- Do explanations have predictive fidelity: does removing/perturbing high-IG features flip predictions (fidelity+), while perturbing low-IG features does not (fidelity−)?
- Please justify max depth=4 and min leaf=40?
- Why is IDS framed as supervised classification rather than an anomaly-first pipeline, and how would the method adapt to zero-day attacks?
- Can you clarify “attribution” and the “2.7× higher” statement: is it the ratio of global mean |IG| of frame.time_relative to the next feature?

---

### Official Review · Reviewer_VRxM · 2025-10-28

**Soundness:** 2
**Presentation:** 2
**Contribution:** 2
**Rating:** 2
**Confidence:** 4

**Summary:**

The paper introduces ExAI5G, an explainable intrusion detection framework designed for 5G network environments. The system combines a Transformer-based deep learning model for traffic classification with logic-based XAI techniques to enhance interpretability. Specifically, the method integrates Integrated Gradients for feature attribution and a surrogate decision tree for rule extraction, generating human-readable logical rules that mirror the model’s decision boundaries. A novel evaluation methodology is also proposed for assessing explanations generated by large language models (LLMs), measuring faithfulness, semantic similarity, and actionability. Empirical results on a 5G IoT intrusion dataset show 99.9% accuracy and a 0.854 macro-F1, along with 16 logical rules achieving 99.7% fidelity to the original model.

**Strengths:**

1. Achieves near-perfect classification accuracy without compromising explainability, a rare combination in IDS systems.

2. The use of both Integrated Gradients (for continuous attribution) and surrogate decision trees (for symbolic reasoning) provides multi-level interpretability feature and rule-level explanations.

3. The proposal to use LLMs as evaluators for explanation quality (faithfulness, semantics, actionability) is creative and potentially generalizable to other XAI domains.

4. The reported 99.7% rule fidelity indicates that the extracted logic closely aligns with the Transformer’s actual reasoning, suggesting strong model transparency.

**Weaknesses:**

1. The IDS backbone is a standard Transformer architecture; the novelty primarily lies in the integration and evaluation pipeline rather than a fundamentally new detection model.

2. The results are limited to a single benchmark (5G IoT intrusion dataset), which may not reflect real network heterogeneity, noise, or adversarial conditions.

3. Relying on another LLM to assess explanation faithfulness could introduce bias or circular validation, especially if both models share similar training data or reasoning styles.

4. It remains uncertain how ExAI5G scales in real-time inference scenarios, where 5G systems may require millisecond-level latency and high throughput.

**Questions:**

1. How do you ensure that the decision tree surrogate remains stable and faithful to the Transformer model when the latter is retrained or fine-tuned on new traffic data?

2. Which LLM model and prompt design were used for evaluating faithfulness and actionability, and how do you quantify inter-LLM reliability or bias in these evaluations?

3. Since interpretability is a key goal, how does ExAI5G behave when encountering novel attack types or traffic signatures unseen during training?

4. Can the system’s logic rules adapt or flag “unknown” cases in a human-understandable manner?

---

### Official Review · Reviewer_NJRc · 2025-10-30

**Soundness:** 2
**Presentation:** 3
**Contribution:** 1
**Rating:** 2
**Confidence:** 4

**Summary:**

This paper presents a pipeline for explaining predictions made by a TabTransformer model trained for intrusion detection tasks. The approach first computes feature attributions using Integrated Gradients (IG), and then trains a decision-tree surrogate model to provide an inherently interpretable approximation of the transformer’s behavior. Finally, both the attribution results and the surrogate model are passed to an LLM to generate high-level textual explanations for end-users. The method is evaluated on a 5G/IoT intrusion detection dataset, demonstrating that the transformer achieves strong accuracy while the pipeline produces corresponding explanations.

**Strengths:**

1. Intrusion detection is an important task, and in certain settings, being able to explain the system’s decisions is particularly valuable.

2. The concept of combining multiple explanation approaches, feature attributions, interpretable surrogate models, and LLM-generated textual explanations, is interesting, although the way it is integrated here feels fairly straightforward.

3. Overall, the paper is clearly written and easy to understand.

**Weaknesses:**

1. **Limited novelty**: The work does not appear to introduce a fundamentally new interpretability or explainability concept. Instead, it combines known components into a pipeline: integrated gradients for attribution, decision trees as surrogate models, and LLM-generated textual explanations based on existing explanations, an idea previously explored, as acknowledged by the authors.

2. **Unclear design choices**: The rationale behind key decisions is not well-justified. For example, why use Integrated Gradients rather than other attribution methods? Why train decision trees instead of alternative interpretable models such as linear models, GAMs, or EBMs?

3. **Questionable need for Transformers**: It is not clearly justified why a Transformer is necessary for this tabular classification task. Tabular benchmarks are widely dominated by simpler models such as tree ensembles (e.g., XGBoost), and interpretable variants like EBMs often achieve comparable performance. The authors present a single model where their Transformer outperforms XGBoost, but this is insufficient to support the general claim that Transformers are needed for tabular intrusion-detection tasks. Is this a common practice supported by prior literature? Is the architecture used here standard for such settings? The choice risks appearing cherry-picked to support the narrative.

4. **Misleading interpretability claim**: The statement that the framework “achieves 99.87% accuracy while maintaining complete interpretability” is misleading. The method does not produce an inherently interpretable model; rather, it applies post-hoc explanations. Reporting accuracy in this context may wrongly imply that the model itself is both high-performing and inherently interpretable, which is not the case.

5. **Domain-specific motivation unclear**: Apart from the chosen dataset being related to intrusion detection, it is not clear why this pipeline is tailored to 5G intrusion-detection tasks. The same pipeline could seemingly be applied to almost any classification problem. What makes this domain unique for the proposed framework?

6. **Limited experimental validation**: Experiments are conducted only on a single dataset. Additional benchmarks are needed to demonstrate generalizability and robustness of the approach.

7. **Lack of ablations**: The paper does not include ablation studies to evaluate the effect of key design and hyperparameter choices.

**Questions:**

1. What makes this pipeline tailored to 5G/IoT intrusion detection specifically? If the approach is meant to be general, why is the paper and the experimental evaluation focused solely on that domain rather than testing broader tasks?

2. What is the justification for using Transformers for tabular intrusion-detection data? Is their superiority for this setting supported by prior work, or should this choice be further motivated and validated?

---

### Official Review · Reviewer_iczN · 2025-11-01

**Soundness:** 3
**Presentation:** 3
**Contribution:** 2
**Rating:** 0
**Confidence:** 3

**Summary:**

The goal of this paper is to develop an AI framework that can be used for intrusion detection. The focus is on explainability and the proposed architecture combines a number of off-the-shelf AI techniques like LLMs and transformers.

**Strengths:**

+ Intrusion detection is an application that can substantially benefit from explainability.
+ Experimental datasets is very practical.

**Weaknesses:**

- This paper is an off-the-shelf application of AI to intrusion detection and, thus, it is more suitable for security conferences like IEEE CNS rather than ICLR. The AI contribution is very limited.
- The proposed framework can be very complex and may not run well on IoT systems.
- There are no comparisons with other AI techniques such as causality that have been used for explainable intrusion detection.
- The use of the LLM seems contrived.

**Questions:**

- Can you provide the substantial contributions to AI that this work provides to justify its relevance to ICLR rather than network security conferences?
- What happens if the LLM is not used in your architecture? What benefits does it give?
- Can you compare your approach with at least three other baselines for explainability like causal learning?
- What is the complexity of your approach and how would you deploy it in a real IoT?
- What security guarantees can your approach provide?
- What happens if the attacker is strategic and can evade your detection?

---

### Note · Authors · 2025-11-28

I have read and agree with the venue's withdrawal policy on behalf of myself and my co-authors.